# Differential p53-Mediated Cellular Responses to DNA-Damaging Therapeutic Agents

**DOI:** 10.3390/ijms222111828

**Published:** 2021-10-31

**Authors:** Lindsey Carlsen, Wafik S. El-Deiry

**Affiliations:** 1Laboratory of Translational Oncology and Experimental Cancer Therapeutics, The Warren Alpert Medical School, Brown University, Providence, RI 02903, USA; lindsey_carlsen@brown.edu; 2The Joint Program in Cancer Biology, Brown University and the Lifespan Health System, Providence, RI 02903, USA; 3Department of Pathology and Laboratory Medicine, The Warren Alpert Medical School, Brown University, Providence, RI 02903, USA; 4Pathobiology Graduate Program, The Warren Alpert Medical School, Brown University, Providence, RI 02903, USA; 5Cancer Center, The Warren Alpert Medical School, Brown University, Providence, RI 02903, USA; 6Department of Medicine, Hematology-Oncology Division, Rhode Island Hospital, Brown University, Providence, RI 02903, USA

**Keywords:** p53, chemotherapy, radiation, target selectivity, DNA damage

## Abstract

The gene *TP53*, which encodes the tumor suppressor protein p53, is mutated in about 50% of cancers. In response to cell stressors like DNA damage and after treatment with DNA-damaging therapeutic agents, p53 acts as a transcription factor to activate subsets of target genes which carry out cell fates such as apoptosis, cell cycle arrest, and DNA repair. Target gene selection by p53 is controlled by a complex regulatory network whose response varies across contexts including treatment type, cell type, and tissue type. The molecular basis of target selection across these contexts is not well understood. Knowledge gained from examining p53 regulatory network profiles across different DNA-damaging agents in different cell types and tissue types may inform logical ways to optimally manipulate the network to encourage p53-mediated tumor suppression and anti-tumor immunity in cancer patients. This may be achieved with combination therapies or with p53-reactivating targeted therapies. Here, we review the basics of the p53 regulatory network in the context of differential responses to DNA-damaging agents; discuss recent efforts to characterize differential p53 responses across treatment types, cell types, and tissue types; and examine the relevance of evaluating these responses in the tumor microenvironment. Finally, we address open questions including the potential relevance of alternative p53 transcriptional functions, p53 transcription-independent functions, and p53-independent functions in the response to DNA-damaging therapeutics.

## 1. Introduction

*TP53* is a tumor suppressor gene that is mutated in about 50% of all cancers [1]. The encoded protein p53 is a transcription factor that is activated by a variety of cell stressors such as DNA damage, oncogene activation, hypoxia, and nutrient deprivation [2,3]. Once activated, p53 binds to the promoter regions of select target genes to activate their transcription [4]. p53 regulates hundreds of target genes and depending on the subset of genes that it activates, mediates different cell fates such as apoptosis, cell cycle arrest, DNA damage repair, inhibition of angiogenesis, and modulation of metabolism [3,5,6,7]. Target gene selection by p53 is regulated based on p53 accumulation and kinetics in the nucleus, p53 post-translational modification (PTM), and p53-DNA binding which is regulated by p53-binding proteins, cofactors, and chromatin remodeling proteins.

p53 regulates hundreds of target genes [8] and is controlled by a complex regulatory network [9], therefore it is not surprising that p53 target gene selection varies across contexts including cell type, tissue type, and treatment type or dose. For example, distinct p53 target induction profiles were observed across the spleen, thymus, and intestine of γ-radiation-treated mice [10,11]; qualitatively different p53 signaling patterns were observed across a panel of twelve cell lines treated with different doses of DNA damage [12]; and treatment of colorectal cancer cells with either 5-fluorouracil, irinotecan (CPT-11), oxaliplatin, or cisplatin revealed drug-specific p53-dependent gene signatures [13]. Investigation of these differential p53 responses to DNA-damaging agents across multiple levels of the p53 regulatory network has revealed that p53 target selectivity may be regulated on different levels depending on context [14,15]. The molecular basis of target selection driving these differential responses and the relative impact of different drug-, cell type-, or tissue type-specific p53 regulatory network profiles on cell fate are not well understood.

Investigating p53 regulatory network profiles across contexts may inform logical ways to manipulate the network to encourage p53-mediated tumor suppression and anti-tumor immunity in cancer patients. Identification of targets genes that are fundamental to the p53 program in specific cell types or tissue types could guide the development of p53-reactivating therapies. Additionally, evaluating differential effects of DNA-damaging agents on the p53 response may inform combination treatments that optimize the induction of p53-mediated apoptosis in cancer cells and/or enhance the p53-mediated anti-tumor immune response.

The relevance of modulating the p53 regulatory network either alone or in the presence of DNA-damaging agents is clear. Many in vivo studies have documented tumor regression in response to wild-type p53 restoration either through genetic means, i.e., in inducible transgenic models [16,17], gene therapy [18,19] or by use of p53 pathway-restoring therapeutic agents [20,21]. Furthermore, wild-type p53 function predicts better outcomes after treatment with conventional chemotherapy or radiation in a majority of clinical studies [22]. Moreover, p53 activation can synergize with DNA-damaging agents to kill cancer cells [23,24,25,26,27]. Several p53-activating or -reactivating compounds are in various stages of clinical trials [28], but currently there are no FDA-approved therapies that target either wild-type or mutant p53.

Here, we review the basics of the p53 regulatory network and how its response may vary after treatment with different DNA-damaging agents. Next, we discuss recent efforts to characterize differential p53 responses to DNA-damaging agents across treatment type, cell type, and tissue type and examine the relevance of evaluating these responses in the tumor microenvironment (TME). Finally, we address open questions including differential drug responses in alternative p53 transcriptional programs, non-transcriptional p53 functions, and p53-independent effects of DNA-damaging agents.

## 2. Factors Mediating Differential p53 Responses to DNA-Damaging Therapeutic Agents

The differential p53 responses to DNA-damaging agents are likely largely mediated at the level of p53 target gene selection and therefore will be the main focus of this review. p53 target gene selection can be regulated at many levels of the p53 regulatory network, including at the level of p53 expression and accumulation in the nucleus, p53 kinetics, specific p53 PTMs, or differential p53-DNA binding which is regulated by other transcription factors, p53-binding proteins, and/or chromatin remodeling proteins (Figure 1).

### 2.1. Nuclear p53 Accumulation & Kinetics

The level of p53 in the nucleus has large influence on p53 target gene selection. Promoters of p53 target genes mediating cell cycle arrest and DNA repair are more likely to have high-affinity binding sites, whereas promoters of target genes involved in apoptosis have relatively lower-affinity binding sites. Therefore, a high level of p53 is needed to activate lethal pathways such as apoptosis while low levels of p53 lead to DNA damage repair or cell cycle arrest [37]. This mechanism of p53 target gene selection is sometimes referred to as the threshold model. Differences in p53 kinetics, independent of maximum p53 level in the nucleus, can also impact target gene selection. For example, pro-apoptotic genes are more sensitive to sustained p53 activity while p21, which largely mediates cell cycle arrest, is sensitive to short pulses of p53 [38,39,40,41]. Not surprisingly, different anti-cancer drugs can induce different levels of p53 and can have different impacts on p53 kinetics, which could impact s p53 target selection and cell fate [41]. It is also likely that different cell types express, degrade, and/or export p53 from the nucleus at different rates, impacting target gene selection. Though nuclear accumulation and p53 kinetics have a large influence on p53 target gene selection, certain cancer cell lines with high levels of p53 do not experience anti-proliferative activity, indicating that other factors also must play a role [42].

### 2.2. Post-Translational Modifications of p53

p53 PTMs include phosphorylation, acetylation, methylation, neddylation, sumoylation and ubiquitination and play roles in p53 stabilization, activation, and target gene selection [43]. Phosphorylation of specific p53 sites plays a particularly critical role in regulating p53-mediated apoptosis. To date, ten kinases and nine p53 phosphorylation sites which activate apoptosis have been identified, whereas other kinases and sites play a role in other p53-mediated cell fates such as cell cycle arrest and DNA damage repair [29]. Acetylation of p53 also plays an important role in determining cell fate [30,36]. Some important p53 PTM sites, the kinase or acetyltransferase responsible, the kinase/acetyltransferase stimulus, and resulting cell fates are outlined in Table 1.

Some of the most critical phosphorylation sites on p53 include serine 15 (S15), S20, and S46. Phosphorylation of S15 and S20 disrupts p53 interaction with its negative regulator MDM2, preventing p53 ubiquitination and degradation. Phosphorylation at these sites also induces apoptosis [31]. Phosphorylation of p53 at S46 plays a particularly crucial role in inducing apoptosis. Phospho-p53(S46) levels correlate with p53-regulated apoptosis-inducing protein 1 (p53AIP1), which promotes apoptosis through the release of mitochondrial cytochrome c [32]. The phosphorylation of p53 at these sites and others is mediated by kinases ATM, ATR, HIPK2, DYRK2, PKCδ, ERKs, CKs, p38 MAPK, among others (Figure 1). These kinases are activated by particular cell stressors. There is much redundancy in this system, as several kinases can be activated by the same type of cellular stressor, many kinases can phosphorylate p53 at different sites, and the same cell fate can be achieved by multiple different p53 PTMs [32]. Therefore, despite the fact that some kinases are reactive to particular stressors (for example, ATM vs. ATR are responsive to UVA vs. UVC light, respectively), the cell same fate may still be achieved [31].

There are several facts that complicate p53 PTM regulation. First, there is some evidence to support lack of redundancy in p53 PTM regulation in terms of specific kinases or PTM sites. For example, there are some sites which may only be phosphorylated by one specific kinase (S6, S9 and T18 by CK1; T81 by JNK) though this finding may be due to the lack of p53 site-specific antibodies or other methods of detecting specific modifications on p53 [31]. Furthermore, some specific sites on p53 are thought to be phosphorylated by specific types of damage. For example, phosphorylation of S389 by p38 kinase following UV irradiation [44,45]. Finally, some p53 PTMs are a prerequisite for PTMs at other sites. For example, CBP-mediated acetylation of p53 at K382, which together with acetylation of K120 and K320 may induce p21-mediated senescence [30], first requires phosphorylation of S46, which induces apoptosis [14,46]. This cross-talk may expand the total number of possible cell fates that can be mediated by any one PTM, however the precise regulation of p53 PTM cross-talk is not well understood [35]. These complicating factors necessitate evaluation of complete PTM profiles over time to thoroughly investigate the relationship between particular DNA-damaging agents, p53 PTMs, and cell fate. This type of evaluation may reveal unique p53 PTM combinations that result in distinct cell fates after treatment with particular agents [14], as the relationship between specific cell stressors, p53 PTM profiles, and target gene selection remains unclear [35].

### 2.3. p53 Binding Proteins, Cofactors, and Chromatin Remodeling

Several factors regulate p53-DNA binding including p53-binding proteins, p53 cofactors, and chromatin remodeling proteins. p53-binding proteins physically interact with p53 and influence its binding to promoters of target genes. For example, when bound to p53, the apoptosis stimulating proteins of p53 (ASPPs) specifically enhance apoptosis instead of cell cycle arrest through activation of BAX. Other p53-binding proteins that selectively stimulate apoptosis include p53β, Brn3b, NFκB/p52, Muc1, and Pin1. p53-binding proteins that can selectively promote cell cycle arrest include Brn3A, Hzf, c-Abl, YB1, p18/Hamlet [32]. More recently, FOXO4 was found to interact with p53 to promote the upregulation of p21 and senescence [33].

p53 cofactors present at the promoter of p53 target genes can also affect p53-mediated transcription activation. These include hCas/p52, which enhances apoptosis, Muc1/Miz and E2A, which enhance cell cycle arrest, SLUG, which inhibits apoptosis, and Zbt4/p52, which inhibit cell cycle arrest [32,34,47]. However, recent investigation has revealed that most often, p53 activation of target genes may not rely on the presence of cofactors at the target gene promoter [48,49]. It is reasonable to believe that the basal level of specific p53-binding proteins and cofactors varies across cell and tissue type, and that levels of p53-binding proteins and cofactors may be regulated differently across drugs used to treat cancer [4]. However, the relative impact of these variations across cell type in the differential p53 response across DNA-damaging agents has not been investigated thoroughly.

Chromatin remodeling is the process by which chromatin architecture is changed to expose certain areas of the genome to allow for transcription. Chromatin remodeling proteins such as histone acetyltransferases, kinases, and methyltransferases modulate acetylation, phosphorylation, and methylation levels of histones, respectively, impacting the availability of specific areas to transcriptional machinery. Acetylation and phosphorylation of histones generally enhance gene transcription, whereas methylation generally represses it, though methylation of some sites can enhance transcription [50]. Several chromatin remodeling proteins regulate availability of p53 target genes for transcription. For example, hCAS/CSE1L associates with select p53 target genes and downregulates methylation of histone H3K27, enhancing transcription of pro-apoptotic p53 target genes [51]. Additionally, methyltransferases PRMT1 and CARM1 cooperate with acetyltransferase p300/CBP to increase transcription of GADD45 after UV irradiation in a p53-dependent manner [4]. Levels of chromatin remodeling proteins may vary across cell type, and thus different cells could mediate cell-type specific induction of p53 target genes and cell fate. For example, p53 binds to the p21 promoter with equal efficiency in human embryonic stem cells and in differentiated mesenchymal stem cells, but p21 transcription is suppressed by H3K27 methylation specifically in embryonic cells [9,52] possibly due to varying levels of chromatin remodeling proteins.

### 2.4. Models of p53 Target Gene Selection Regulation

The above-mentioned regulatory mechanisms have been considered in a series of proposed models describing p53 target selection, including the selective binding model and selective context model. In the selective binding model, target selectivity is driven by different DNA binding profiles. On the other hand, the selective context model suggests that p53 binds to all sites of the genome that are accessible, then factors including p53 PTMs and p53 cofactors determine the final cellular outcome [14]. It is likely that no one model will hold true across all contexts and that a complex interplay of p53 regulatory mechanisms mediate p53 target selection upon particular cell stressors [52]. Investigating what mechanisms are important in particular contexts and unraveling p53 regulatory mechanism interplay is essential to allow clinical manipulation of the p53 program in a way that encourages p53-mediated tumor suppression or anti-cancer immunity in cancer patients.

In addition to considering all aspects of the p53 regulatory network, models of p53 target selection regulation should also recognize factors that do not impact p53 directly, but modulate cell fate through other mechanisms. For example, ATM signaling protects cells from p53-mediated apoptosis not by regulating p53-mediated target gene transcription, but by blocking autophagy which helps to maintain mitochondrial homeostasis and suppress ROS levels [41].

## 3. Ongoing Efforts to Characterize Differential p53 Responses to DNA-Damaging Therapeutic Agents

The regulation of p53 target gene selection based on the presence of specific p53 PTMs and level of nuclear p53 accumulation, binding proteins, cofactors, and chromatin modifications is well-characterized. However, the complex interplay of these processes across various drug treatments, their cumulative effect in terms of ultimate cell fate determination, and how they can be clinically exploited remain difficult tasks to pursue. Several factors contribute to this challenge, but possibly the most important is the necessity to evaluate (1) the expression level of p53, (2) p53 PTMs, (3) p53-DNA binding profiles, (4) levels of chromatin remodeling proteins and (5) transcription and translation of p53 target genes simultaneously in order to fully investigate the mechanism by which different drugs mediate different p53 responses. The persistence of damage [53], the status and activity of various DNA repair pathways [12,54], the kinetics of the p53 response [38,39,40,41], and cellular heterogeneity [12,15,41,52] also impact on the various observed effects on p53 and its function. It is often not possible to investigate each of these levels of regulation simultaneously due to the time-consuming and costly processes they involve. Nonetheless, some progress has been made as far as investigating multiple levels of p53 target gene regulation across multiple types of DNA-damaging agents, cell types, and tissue types. The following sections discuss this progress and highlight the future work that could aid in the further clinical exploitation of differential p53 responses.

### 3.1. Efforts to Establish a Context-Independent p53 Program

Some investigators have evaluated many datasets across different cell types and treatments to identify genes that are regulated in a context-dependent or -independent manner. These analyses have provided evidence to suggest that functional p53 binding is highly conserved across different experimental conditions including cell type and treatment type, and that differences across experiments can largely be attributed to nonfunctional binding events [48]. Furthermore, a meta-analysis of 20 genome-wide p53 gene expression profiles and 15 p53 binding profiles showed that many p53 target genes are regulated across cell type and treatment type [55]. Finally, another recent analysis found that only two genes (CDKN1A and RRM2B) were commonly identified as p53 targets across 16 datasets, which was striking because some datasets were established using the same treatment conditions. This may speak to the lack of reproducibility across individual experiments or prevalence of false positives in these types of analyses [49]. Despite these results, investigation across a wide variety of cell types and treatment types may still reveal context-dependent p53 regulatory mechanisms that can be specifically targeted based on cancer type and treatment type. Accordingly, the studies discussed below suggest that DNA-damaging therapeutic agents do in fact induce differential p53 responses.

### 3.2. Differential p53-Mediated Cellular Responses to DNA-Damaging Therapeutic Agents

Few studies have treated the same cell type with multiple different DNA-damaging agents and evaluated the p53 response. Though limited in number, each investigation has made important observations supporting the idea that there is variation in p53 signaling and/or cell fate across treatment type that could be targeted to enhance the p53-mediated apoptotic response.

We recently published a study that directly compared the p53 transcriptional response in HCT116 colorectal cancer cells treated with 5-FU, irinotecan, oxaliplatin, or cisplatin at equitoxic doses [13]. Multiple types of signatures were established including p53-dependent, p53-independent, pan-drug, drug-specific, and drug class-specific. Drug-specific p53-dependent transcripts included upregulation of toll-like receptor 3 (TLR3) by irinotecan, suggesting irinotecan-specific promotion of anti-cancer immunity through activation of type I interferon (IFN) [56]. SAT1, which plays a critical role in ferroptosis, was upregulated in a p53-dependent, oxaliplatin-specific manner which suggests an oxaliplatin-specific role in this mode of cell death in colorectal cancer cells [57]. Despite the majority of the response to the drugs being drug-specific, several genes were regulated across 5-FU, irinotecan, oxaliplatin, and cisplatin including BTG2, suggesting a critical role of this DNA damage response protein in the cellular response to chemotherapy in colorectal cancer [13,58]. Many questions remain, including why differences were observed across the similar platinum-based drugs cisplatin and oxaliplatin, and what part of the p53 regulatory network mediates these differential responses.

Another study investigated differential responses of p53 after treatment of U2OS osteosarcoma cells with either Actinomycin D or Etoposide, two commonly used DNA-damaging agents in cancer treatment. Actinomycin D treatment resulted in more growth arrested cells, whereas Etoposide tended to induce apoptosis [14]. Further investigation revealed the drugs induced different levels of p53 S46 phosphorylation, very similar genome-wide p53-DNA binding patterns, and treatment-specific p53 target gene expression patterns. This study clearly demonstrated enhancement of apoptosis after Etoposide treatment via increased functional phospho-p53(S46)-DNA binding. However, phospho-p53(S46) was found to be bound at other sites in addition to apoptotic target genes, and DNA binding did not correlate globally with gene expression. Together, these results suggest that there are other important layers of regulation needed to mediate apoptosis by Etoposide, and that apoptosis may not be the sole functional role of phospho-p53(S46). Understanding how these mechanisms function to mediate apoptosis may help us understand how to direct the p53 program in a similar way with other drugs in other cancer types.

A similar study was performed on MCF7 breast cancer cells treated with three p53 activators that vary in mechanism: nutlin3a, RITA, and 5-FU. Nutlin3a inactivates MDM2, a negative regulator of p53; RITA directly binds p53 and enhances its expression [59]; and 5-FU incorporates into DNA and inhibits thymidine synthase, causing altered nucleotide pools and DNA damage which activates p53 [60]. Growth arrest was observed in nutlin3a and 5-FU-treated cells and apoptosis was observed in RITA-treated cells. Despite mediating different cell fates, the drug treatments induced similar p53-DNA binding profiles. Further investigation revealed that the transcription factor Sp1 plays a major role in the RITA-induced p53 program, as depletion of Sp1 converted the RITA-induced expression profile to a nutlin3a-like pattern [59]. It is possible that Sp1 overexpression with 5-FU or nutlin3a treatment may enhance p53-mediated apoptosis in the context of breast cancer. This study serves as a robust example of p53-activating target discovery via thorough investigation of the p53 regulatory network.

### 3.3. Differential p53-Mediated Cellular Responses to DNA-Damaging Therapeutics across Cell Type

Differential drug responses of p53 are also observed across cell type and cell line. For example, one study found that across twelve different cell lines, the basal expression of p53 was comparable but p53 dynamics after treatment with radiomimetic drug neocarzinostatin varied substantially, even across some cell lines of the same tissue type [12]. Further investigation revealed that different cell types have different baseline ATM activity and DNA repair efficiencies, and depending on these baseline levels, cells experience qualitatively different p53 signaling patterns in response to different doses of DNA damage. For example, A549 lung cancer cells have high baseline ATM activity and moderate DNA repair efficiency, and when treated with high doses of DNA-damaging agents are pushed out of oscillatory p53 signaling and into more sustained p53 activation. On the other hand, MCF7 breast cancer cells have low baseline ATM activity, which causes them to remain in the oscillatory pattern even when treated with high doses of DNA-damaging agents. Measurement of baseline levels of ATM and DNA repair efficiency across cancer types may reveal rational combination therapies to achieve specific cellular signaling dynamics in any given tumor. Though no direct readout of cell fate was measured, this study identified ATM kinase as a potential target to modulate p53 signaling and again demonstrates the potential for these types of studies to identify clinically relevant targets.

Another study investigated HCT116 colorectal cancer cells and IMR90 normal fibroblasts treated with 5-FU and found variation in their p53-DNA binding profiles. This finding is likely due to cancer-associated epigenetic changes found in HCT116 cells [15]. Differences in epigenetic state may explain cell type-specific p53 target induction in many cases, such as in human embryonic stem cells, which suppress p21 transcription via cell type-specific H3K27 methylation [41,52]. Identification of the specific epigenetic changes that mediate p53-DNA binding at apoptotic target genes, and correlation of these epigenetic changes with particular cell fates, could direct the development of dual treatment with chromatin remodeling and p53-activating compounds.

### 3.4. Tissue Specificity in the p53 Response to DNA-Damaging Therapeutic Agents

In addition to differential p53 responses to DNA-damaging agents across treatment type and cell type, the p53 response also varies across tissue type. For example, in mice treated with γ-irradiation, apoptotic p53 target genes DR5, Bid, Puma, and Noxa were upregulated in the jejunum and ileum, whereas p21 was upregulated in the liver and this corresponded with relative increased and decreased sensitivity to irradiation, respectively [61]. Similar experiments evaluated p53 targets in the spleen, thymus, and small intestine and demonstrated that distinct targets may mediate γ-irradiation-induced apoptosis across these tissues and that a “group effect” of several p53 targets is important for p53-dependent apoptosis in some tissues whereas others rely on only one or a few [10,11].

A recent study evaluated the p53 response across pancreas, small intestine, ovary, kidney, and heart via deletion of MDM2 and found a seven-gene signature that was regulated across tissues in addition to a large list of tissue-specific p53 target genes [62]. Another investigation revealed that p53 dynamics vary across tissues, resulting in differential sensitivity to irradiation. Specifically, tissues with transient radiation-induced p53 activation such as the small and large intestines are relatively resistant to irradiation and sustained p53 activation with an MDM2 inhibitor improved response in these tissues [63]. Similarly to cell type-specific differences, tissue-specific differences in p53 target gene profiles may be due to variation in the availability of p53 DNA response elements (REs) as a result of different chromatin modification states across cell and tissue type [9,64].

While several of the more recent examples provided here use p53-reactivating compounds to evaluate differential p53 responses, it is likely that treatment with DNA-damaging agents would also reveal tissue-specific effects. These studies confirm tissue specificity in the p53 response which may inform vastly different optimal combination therapies depending on cancer type.

### 3.5. Differential p53-Mediated Cellular Responses to DNA-Damaging Therapeutic Agents in the Tumor Microenvironment

The TME includes blood vessels, immune cells, cancer-associated fibroblasts (CAFs), signaling molecules including cytokines and chemokines, and the extracellular matrix (ECM) that surround the tumor [65]. Components of the TME are known to influence tumor development, progression, and drug resistance and regulate immune responses in the tumor [66]. The upregulation of p53 in the TME promotes anti-tumor immunity (Figure 2) [67], but the relative contribution of different DNA-damaging agents to these p53-mediated responses is poorly understood. Further investigation is important as the contributions of p53-dependent and -independent immunogenic cell death (ICD) and anti-tumor immunity to the efficacy of DNA-damaging agents are increasingly appreciated [68,69,70]. Moreover, in vivo studies clearly demonstrate the relevance of studying p53-mediated anti-tumor immunity. The p53-reactivating compound APG-115 demonstrated synergy with immune checkpoint blockade [71], local activation of p53 in the TME enhances antitumor immunity through ICD and p53-dependent elimination of immunosuppressive MDSCs [72], and NK cells were shown to be crucial for efficacy of the p53-activating drug DS-5272 in AML [73].

The upregulation of p53 in the TME promotes both innate and adaptive anti-tumor immunity [67]. In the context of innate immunity, treatment with DNA-damaging agents and IFN stimulates p53 interaction with IFN regulatory factors (IRFs), particularly IRF-5 and IRF-9 [67]. Whereas IRF-5 contributes to the anti-cancer response as a regulator of cell growth and apoptosis [74,75], IRF-9 may contribute to survival of cancer cells through upregulation of IL-6 [76], indicating that activation of p53 in the TME should be completed via rational combination therapies to avoid potential pro-cancer effects.

p53 can directly enhance the expression of anti-tumor TLRs such as TLR3 in cancer cells and lymphocytes [77,78,79]. TLR3 can play an anti-cancer role by regulating downstream signaling that enhances expression of pro-inflammatory cytokines, chemokines, and interferons [56]. Our lab’s recent investigation revealed unique, p53-dependent upregulation of TLR3 by irinotecan compared to treatment with 5-FU, oxaliplatin, or cisplatin at equitoxic doses in colorectal cancer cells, suggesting a novel irinotecan-specific impact on the TME [13]. Furthermore, earlier studies demonstrate that the induction of TLRs varied across treatment with the DNA-damaging agents doxorubicin, 5-FU, ionizing radiation, and UV irradiation as well as across different cell lines [78]. Despite recent failure of TLR agonists in the clinic [80] and recent work suggesting a possible contribution of TLR3 to tumor progression [56], TLR-based therapies combined with different types of DNA-damaging agents are currently in clinical trials for treatment of many cancer types [81,82]. It will be interesting to evaluate if effects on TLR expression by DNA-damaging agents impact the ability of TLR agonists to provide additive or synergistic benefit when these treatments are combined.

In the TME, p53 can also modulate levels of cell cycle regulatory genes, growth suppressive genes, apoptosis-inducing genes such as caspases and TRAIL, and immunomodulatory genes such as MHC-I [83] and ISG-15 [84]. Additionally, DNA-damaging agents [85] or p53 activation by other means [86,87,88] can upregulate the NK cell ligand ULBP2 to stimulate NK cell anti-tumor activity, however comparison across drugs has not been investigated. In the context of adaptive immunity, p53 plays a critical role in integrating growth signals that select for activated effector T cells [89].

Validation of most of the mechanisms described above is needed. Further investigation of the impact of different DNA-damaging agents on the TME may inform potential combination therapies to optimize anti-tumor immune effects in patients receiving these therapies. One challenge involving the pharmacological activation of p53 to induce anti-tumor immunity is immune cells’ heightened sensitivity p53 activation compared to tumor cells [65,90]. However, unlike with p53-mediated cytotoxic therapies, p53-mediated immunostimulatory therapies only need to reach part of the tumor, after which the activated immune cells will circulate. Carefully designed dosing and delivery is necessary, but nonetheless this treatment approach may induce long-lasting tumor suppression through activation of the innate and adaptive immune systems [65].

## 4. Conclusions and Open Questions

It is clear that differential p53 responses to DNA-damaging agents will require further preclinical investigation prior to more effective clinical exploitation. Further study may identify context-independent elements of the p53 regulatory network that are needed for apoptosis or may reveal context-specific p53 responses that may be modulated or enhanced with combination treatments. This may involve dual treatment with multiple DNA-damaging agents which have the cumulative desired p53-dependent effect in the tissue of interest or may involve combination of DNA-damaging agents with p53-modulating compounds which show promising synergy in preclinical studies [23,24,25,26,27].

Open questions remain regarding the p53 regulatory network and its relevance to tumor suppression. For example, the most important targets for tumor suppression may vary across cancer type and the identity of these targets remain ill-defined [91,92]. Thus, while identification of context-independent elements of the p53 regulatory network could inform the development of p53-reactivting therapies, it remains to be determined if these targets will be biologically relevant to tumor suppression across several cancer types [93]. Moreover, the contribution of p53 to tumor suppression in general is still unclear. p53-mediated cell cycle arrest and apoptosis in response to DNA damage have been suggested to not be necessary for tumor suppression in some cancer types [92]. This raises the question of alternative p53 transcriptional programs including ferroptosis, regulation of stemness, and metabolism [8] which may also be induced in a drug-specific manner. In fact, SAT1, which contributes to p53-mediated ferroptosis [57], was uniquely upregulated by oxaliplatin compared to treatment with 5-FU, irinotecan, or cisplatin in colorectal cancer cells [13] which indicates a potential unique role of oxaliplatin in this mode of cell death that is distinct from classical p53-mediated apoptosis. Also recognized are non-transcriptional functions of p53 that affect a wide range of cellular processes such as apoptosis, growth suppression, DNA repair [94,95]. These transcription-independent mechanisms are less extensively studied as compared to p53 transcriptional activity, especially in terms of differential p53 responses to DNA-damaging agents. However, they may be relevant, as apoptosis can still occur in cells with transcriptionally dysfunctional p53 [94]. Further investigation is needed as the development of p53-reactivating/activating therapies continues.

The relative contribution of p53-independent effects of DNA-damaging agents cannot be ignored as they also play major roles in the cellular response to therapy. For example, the primarily pro-survival integrated stress response (ISR) can mediate resistance to chemotherapy. Similarly to p53, the cell fate mediated by the ISR depends on the extent of damage, and thus differential responses across different DNA-damaging agents is likely [96]. Other important components of the p53-independent response to chemotherapy are p63 and p73, which are members of the p53 family of transcription factors. By contrast to p53, p63 and p73 are rarely altered in cancer and are most commonly expressed as N-terminally truncated isoforms [97]. p63 and p73 dysregulation contributes to tumor promotion both through p53-dependent and p53-independent mechanisms, including possible p53-indepdenent regulation of adhesive signaling, Notch activation, and Rb phosphorylation by p63 [98,99,100,101] and induction of Puma-mediated apoptosis by p73 [102]. Furthermore, it is important to consider the impact of p53 in counteracting therapeutic cellular responses to DNA-damaging agents. Our recent work identified a p53−/− cell-specific, oxaliplatin treatment-specific upregulation of IL-8 and a p53−/− cell-specific, cisplatin treatment-specific upregulation of ferritin among colorectal cancer cells treated with either 5-FU, irinotecan, oxaliplatin, or cisplatin. These findings further support the investigation of p53-independent drug effects of such pro-tumorigenic cytokines [13].

Simultaneous evaluation of the p53 regulatory network across different treatment types, cell types, and tissue types is a major and daunting task, especially when taking into consideration alternative p53 transcriptional function and transcription-independent effects. Furthermore, it is important to consider the relationship between direct modification of p53 by DNA-damaging agents and the alteration of cellular context by these agents. Various aspects of cellular context may be influenced by drug treatment, impacting not only p53-dependent responses but also p53-independent responses that impact on cell fate. Careful consideration of this relationship should be taken when evaluating differences in p53-mediated cell fate across drugs. The breadth of work that is required to fully understand the p53 response may explain why the molecular basis of p53 target gene selection across these contexts is not well understood. Despite this challenge, the knowledge gained from examining specific p53 regulatory network profiles across contexts may help to move clinical practice away from empirical decision making and toward logical combinations based on established molecular mechanisms of specific agents in particular tissues.

## Figures and Tables

**Figure 1 ijms-22-11828-f001:**
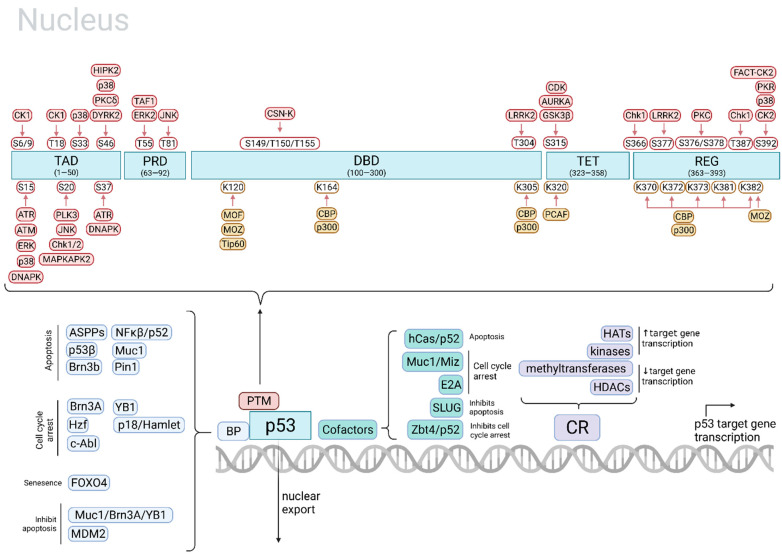
Regulation of p53 target gene selection. Top: p53 PTM plays a major role in regulating p53 target gene selection. p53 PTMs include phosphorylation by kinases (red) and acetylation by acetyltransferases (yellow). The cell fates mediated by PTM of specific sites by particular kinases/acetyltransferases are outlined in Table 1. Bottom: In addition to p53 PTMs, p53 target gene selection is also regulated by the rate of p53 transcription and nuclear export, p53 kinetics, p53-binding proteins, p53 cofactors, and chromatin remodeling proteins. TAD, p53 transactivation domain; PRD, proline-rich domain; DBD, DNA-binding domain; TET, tetramerization domain; REG, C-terminal regulatory domain; BP, binding protein; PTM, post-translational modification; CR, chromatin remodeling protein; HAT, histone acetyltransferase; HDAC, histone deacetylase. Adapted from [29,30,31,32,33,34]. Created in BioRender.

**Figure 2 ijms-22-11828-f002:**
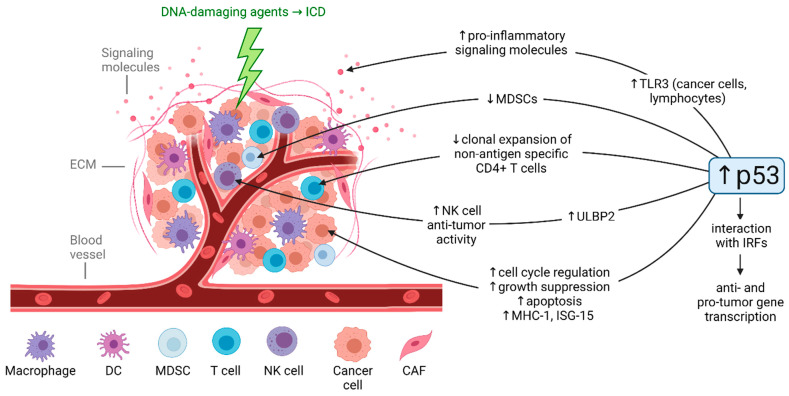
Effects of p53 activation on the tumor microenvironment. The TME includes blood vessels, immune cells, CAFs, signaling molecules including cytokines and chemokines, and the ECM that surround the tumor. DNA-damaging agents induce ICD, which induces innate and adaptive anti-tumor immune responses. The development of these anti-tumor immune responses is partially regulated by p53, which can upregulate anti-tumor TLRs such as TLR3, downregulate immunosuppressive MDSCs, enhance antigen-specific CD4+ T cell clonal expansion, upregulate NK cell ligand ULBP2 which enhances NK cell anti-tumor activity, upregulate cell cycle regulatory, growth suppressive, and apoptosis-inducing genes as well as immunomodulatory genes MHC-I and ISG-15, and interact with IRFs leading to stimulation of the anti-cancer (but also pro-tumor) response. Created in BioRender. DC, dendritic cell; MDSC, myeloid-derived suppressor cell; CAF, cancer-associated fibroblast; ECM, extracellular matrix; ICD, immunogenic cell death; TLR, toll-like receptor; IRF, interferon regulatory factor; MHC, major histocompatibility complex; ISG, interferon-stimulated gene.

**Table 1 ijms-22-11828-t001:** Examples of p53 post-translational modifications and the cell fates they mediate. Various p53-mediated cell fates including apoptosis, cell cycle arrest, DNA repair, senescence, and ferroptosis are achieved by PTM of p53 at specific sites. PTM is carried out by various kinases or acetyltransferases, which are stimulated by distinct stimuli. The main target genes mediating these cell fates are listed if known. PTM, post-translational modification; ↑ increase or increased expression; ↓ decrease or decreased expression. Adapted from [30,31,32,35].

Main Cell Fate	p53 Site and PTM	p53 Modifier	Stimulus	Main Target Gene(s)	Ref.
Apoptosis	phospho-S15	ERKs	UV light		[31]
	phospho-S15	P38	UV light		[31]
	phospho-S15, -S37	ATR	γ-radiation, UV light		[31]
	phospho-S20	JNK	UV light		[31]
	phospho-S20	MAPKAP2	UV light		[31]
	phospho-S46	HIPK2	UV light	e.g., ↑ AIP1	[31,32,35]
	phospho-S15	ATM	DNA damage		[31]
	acetyl-K120, -C-terminal (concurrent phospho-S46 needed)	Tip60, MOF, p300/CBP, PCAF	DNA damage, other genotoxic stresses	↑ Bax, Fas, Noxa and Puma	[30]
	acetyl-K120	hMOF, Tip60		e.g., ↑ Puma	[32,35]
Stabilization; apoptosis	phospho-S33, -S46	p38	UV light		[31]
Apoptosis, cell cycle arrest	phospho-S46	HIPK2	UV light		[31]
Cell cycle arrest, apoptosis	acetyl-K164	p300, CBP		Likely important for the activation of the majority of p53 target genes	[32,35]
Cell cycle arrest, DNA repair	acetyl-C-terminal (concurrent phospho-N-terminal needed)	p300/CBP, PCAF; binding by Tip60 w/o acetylation	DNA damage, other genotoxic stresses	↑ p21, GADD45 ↓ Noxa, Pidd	[30]
Cell cycle arrest, promotes cell survival	acetyl-K320	PCAF		↑ p21	[32,35]
Senescence	acetyl-K120, K320, K382 (concurrent phospho-S15, -S20 needed)	MOZ, PCAF, p300	DNA damage, oncogene activation	↑ p21	[30]
Ferroptosis	acetyl-K101	CBP			[35,36]
↑ p53 transcription	phospho-S315	CDK (CDC2/CDK2)	UV light		[31]
↑ p53 activity	phospho-S392	FACT-CK2	UV light		[31]
	phospho-T55	ERK2	Doxorubicin		[31]
Stabilization	phospho-T81	JNK	DNA damage		[31]
	phospho-S6, -S9, -T18 (concurrent phospho-S15 needed)	CK1	Topoisomerase-directed drugs and DNA damage	↓ MDM2	[31]
	phospho-S20	Chk1/2	Ionizing radiation		[31]
	phospho-S15, -S37	DNAPK	DNA damage		[31]
↑ DNA-binding activity of p53	phospho-S392	P38	UV light, DNA damage		[31]
Ubiquitination and degradation; ↑DNA-binding affinity	phospho-S376, -S378	PKC	Unstressed state; constitutively phosphorylated and dephosphorylated with IR light		[31]
Degradation	phospho-T150, -T155, -S149	CSN-associated kinase complex	Unstressed state		[31]
Degradation or stabilization of p53	phospho-T55	TAF1	Constitutively phosphorylated		[31]
↓ p53-mediated apoptosis	phospho-S315, S376	GSK3β	Endoplasmic reticulum stress		[31]

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
