# Peer review of "Differential p53-Mediated Cellular Responses to DNA-Damaging Therapeutic Agents"

_ijms, 2021, doi:10.3390/ijms222111828_

Round 1
Reviewer 1 Report
Carlsen and El-Deiry have constructed a well written review outlining the differences in p53 responses to DNA damaging agents. In addition to summarizing the current literature of p53 responses, they give perspectives on the field, provide logical directions that the field is likely to take and address some of the open questions in the field. Overall, it is important for the audience to understand the current state of the field and this review serves that purpose very well. I recommend accepting the manuscript with only minor changes as outlined below.
- The authors should check the entire article and provide appropriate citations. E.g. the first line reads “TP53 is a tumor suppressor gene that is mutated in about 50% of all cancers” but does not have any citation. There are several such statements mentioned throughout the article that have been built on previous studies but lack the appropriate reference.
- It is also recommended to amend the introduction section a bit. There are some discontinuities in the intro sections. The second sub-section has important information but does not quite fit the message there. The authors may consider moving it. Additionally, the sub-headings for the intro section are not necessary and the authors may consider removing them.
- Table 2 shows examples of cell fate based on cell type, DNA damaging agent, PTM and target genes. The third row in the table does not really fit in it as there are NA values for PTM and DNA damaging agent. In response to that, the entire table does not serve any specific purpose and does not help the review. I suggest significant additions to the table with several more number of cell lines with the appropriate information on the respective columns or removing the table entirely.
- Section 2.3 on page 4: The heading is p53-DNA binding but the section only focusses on p53 binding to other proteins but very little about DNA binding by p53. The authors may consider overhauling this section by adding the right information about how p53 DNA binding affects the response to different drugs.
- In Fig 1., MDM2 is a major binding partner of p53 and it needs to included in the figure.
- In fig.2, please check the spelling of “signaling”.
- Page 8, line 370: “….efficacy of DNA damaging are increasingly….”. The word agent is missing.
- Section 4 is missing or the last section is mislabelled as section 5.
Author Response
Reviewer 1
Carlsen and El-Deiry have constructed a well written review outlining the differences in p53 responses to DNA damaging agents. In addition to summarizing the current literature of p53 responses, they give perspectives on the field, provide logical directions that the field is likely to take and address some of the open questions in the field. Overall, it is important for the audience to understand the current state of the field and this review serves that purpose very well. I recommend accepting the manuscript with only minor changes as outlined below.
- The authors should check the entire article and provide appropriate citations. E.g. the first line reads “TP53 is a tumor suppressor gene that is mutated in about 50% of all cancers” but does not have any citation. There are several such statements mentioned throughout the article that have been built on previous studies but lack the appropriate reference.
Relevant citations were added. - It is also recommended to amend the introduction section a bit. There are some discontinuities in the intro sections. The second sub-section has important information but does not quite fit the message there. The authors may consider moving it.
The second paragraph in the introduction was moved and is now the fourth paragraph.
Additionally, the sub-headings for the intro section are not necessary and the authors may consider removing them.
Sub-headings were removed. - Table 2 shows examples of cell fate based on cell type, DNA damaging agent, PTM and target genes. The third row in the table does not really fit in it as there are NA values for PTM and DNA damaging agent. In response to that, the entire table does not serve any specific purpose and does not help the review. I suggest significant additions to the table with several more number of cell lines with the appropriate information on the respective columns or removing the table entirely.
Table 2 was removed. - Section 2.3 on page 4: The heading is p53-DNA binding but the section only focusses on p53 binding to other proteins but very little about DNA binding by p53. The authors may consider overhauling this section by adding the right information about how p53 DNA binding affects the response to different drugs.
The title of this section was changed to “p53 binding proteins, cofactors, and chromatin remodeling.” - In Fig 1., MDM2 is a major binding partner of p53 and it needs to included in the figure.
MDM2 was added. - In fig.2, please check the spelling of “signaling”.
Spelling was corrected. - Page 8, line 370: “….efficacy of DNA damaging are increasingly….”. The word agent is missing.
“Agent” was added. - Section 4 is missing or the last section is mislabelled as section 5.
Mislabeling was corrected.

Reviewer 2 Report
“Differential p53 responses to DNA damaging therapeutic agents” by Carlsen and El Deiry provides a well written overview of the multifaceted functions of p53 with appropriate acknowledgement of challenges in the field and work in progress.
The outline of the review helps make this article appropriate for a larger scientific audience (in addition to p53 experts) while maintaining focus on context dependent variation in p53 responses. Despite intense focus on the function of this protein over the past 30 years the article acknowledges ambiguity that persists in p53 function. There have been continued efforts to exploit differences in p53 expression/function in cancer cells for therapeutics which remains a critical area of research interest for a large pool of cancer researchers.
Figures are represented with clarity and provides substantial information which will increase citations of this review article.
Revision for minor spelling mistakes such as line 272 will need to be performed.
Author Response
“Differential p53 responses to DNA damaging therapeutic agents” by Carlsen and El Deiry provides a well written overview of the multifaceted functions of p53 with appropriate acknowledgement of challenges in the field and work in progress.
The outline of the review helps make this article appropriate for a larger scientific audience (in addition to p53 experts) while maintaining focus on context dependent variation in p53 responses. Despite intense focus on the function of this protein over the past 30 years the article acknowledges ambiguity that persists in p53 function. There have been continued efforts to exploit differences in p53 expression/function in cancer cells for therapeutics which remains a critical area of research interest for a large pool of cancer researchers.
Figures are represented with clarity and provides substantial information which will increase citations of this review article.
Revision for minor spelling mistakes such as line 272 will need to be performed.
Spelling mistakes were corrected.

Reviewer 3 Report
This is a well-written, scientifically sound manuscript and I support its publication in IJMS. Moreover, it well complements El-Deiry’s recent review (2021 Aug;1876(1):188556). The only problem I have with it is rather semantic. “Differential responses of p53 to DNA-damaging therapeutic agents” – do the authors really considered a direct interaction of drugs with p53? Surely, such interaction was an important element of the final output (response), but it was likely not the main factor in the reaction of the cell to DNA-damaging agents. Of course, I do not suggest that the authors should solely consider different cellular responses in dependence on the kind of DNA damage, as all aspects of cellular context that may be influenced by a drug should be taken into account. For instance, the authors write about differences across similar Pt-based drugs, which nicely illustrates the complex relationship external factor (e.g., drug) – DNA damage – p53 –– cellular context – cell fate.
I would:
modify the title: “Differential p53-mediated cellular response…”
add 1-2 sentences do the concluding section about a relationship between direct modification of p53 by a DNA-damaging agent and cellular context altered by that agent.
Author Response
This is a well-written, scientifically sound manuscript and I support its publication in IJMS. Moreover, it well complements El-Deiry’s recent review (2021 Aug;1876(1):188556).
The only problem I have with it is rather semantic. “Differential responses of p53 to DNA-damaging therapeutic agents” – do the authors really considered a direct interaction of drugs with p53? Surely, such interaction was an important element of the final output (response), but it was likely not the main factor in the reaction of the cell to DNA-damaging agents. Of course, I do not suggest that the authors should solely consider different cellular responses in dependence on the kind of DNA damage, as all aspects of cellular context that may be influenced by a drug should be taken into account. For instance, the authors write about differences across similar Pt-based drugs, which nicely illustrates the complex relationship external factor (e.g., drug) – DNA damage – p53 –– cellular context – cell fate.
I would:
modify the title: “Differential p53-mediated cellular response…”
Title was modified from “Differential p53 Responses to DNA-Damaging Therapeutic Agents” to “Differential p53-mediated Cellular Responses to DNA-Damaging Therapeutic Agents.”
Subheadings 3.2, 3.3, and 3.5 were edited similarly.
add 1-2 sentences do the concluding section about a relationship between direct modification of p53 by a DNA-damaging agent and cellular context altered by that agent.
The following was added to the last paragraph of the conclusion:
“Also important to consider is the relationship between direct modification of p53 by DNA-damaging agents and the alteration of cellular context by these agents. Various aspects of cellular context may be influenced by drug treatment, impacting not only p53-dependent responses but also p53-independent responses that impact on cell fate. Careful consideration of this relationship should be taken when evaluating differences in p53-mediated cell fate across drugs.”
Other changes were made:
- The order of Table 1 was edited so that it is in order by cell fate.
Minor edits or additions to Figures 1 and 2.
